# Immunogenicity and Safety of Modified Vaccinia Ankara (MVA) Vaccine—A Systematic Review and Meta-Analysis of Randomized Controlled Trials

**DOI:** 10.3390/vaccines11091410

**Published:** 2023-08-24

**Authors:** Lior Nave, Ili Margalit, Noam Tau, Ido Cohen, Dana Yelin, Florian Lienert, Dafna Yahav

**Affiliations:** 1Internal Medicine E, Sheba Medical Center, Ramat-Gan 52621, Israel; cheshcat@gmail.com (L.N.);; 2Faculty of Medicine, Tel Aviv University, Ramat-Aviv, Tel Aviv 69978, Israel; ilimargalit@gmail.com (I.M.);; 3Infectious Diseases Unit, Sheba Medical Center, Ramat-Gan 52621, Israel; 4Department of Diagnostic Imaging, Sheba Medical Center, Ramat-Gan 52621, Israel; 5Bavarian Nordic AG, CH-6301 Zug, Switzerland

**Keywords:** mpox, Modified Vaccinia Ankara, Vaccinia virus, smallpox, vaccines

## Abstract

Prevention of mpox has become an important public health interest. We aimed to evaluate the safety and immunogenicity of the Modified Vaccinia Ankara (MVA) vaccine. We conducted a systematic review and meta-analysis of randomized-controlled trials (RCTs) comparing MVA versus no intervention, placebo, or another vaccine. Outcomes included safety and immunogenicity outcomes. We also performed a systematic review of RCTs evaluating various MVA regimens. Fifteen publications were included in the quantitative meta-analysis. All but one (ACAM2000) compared MVA with placebo. We found that cardiovascular adverse events following two MVA doses were significantly more common compared to placebo (relative risk [RR] 4.07, 95% confidence interval [CI] 1.10–15.10), though serious adverse events (SAEs) were not significantly different. Following a single MVA dose, no difference was demonstrated in any adverse event outcomes. Seroconversion rates were significantly higher compared with placebo after a single or two doses. None of the RCTs evaluated clinical effectiveness in preventing mpox. This meta-analysis provides reassuring results concerning the immunogenicity and safety of MVA. Further studies are needed to confirm the immunogenicity of a single dose and its clinical effectiveness. A single vaccine dose may be considered according to vaccine availability, with preference for two doses.

## 1. Introduction

The recent COVID-19 and mpox outbreaks created a rising interest in strategies for preventing and treating viral diseases [1,2]. While the use of drugs directed at preventing viral replication has proven to be of somewhat limited potency, vaccination is an effective way to reduce morbidity. The use of vaccinia-based vaccines has proven to be a vital tool in the context of smallpox prevention, but not without adverse reactions [3].

Two vaccine types are designed for the prevention of orthopoxvirus infections. The older vaccines are replication-competent vaccines. These include the former Dryvax vaccine and the currently available ACAM2000 vaccine. Serious adverse events have been described with these vaccines, including acute vaccinia syndrome, postvaccinial encephalitis, progressive vaccinia, eczema vaccinatum, and generalized vaccinia. Cardiac complications and transmission to household contacts were also reported with these vaccines [4].

Modified Vaccinia Ankara (MVA) is an attenuated replication-deficient *poxvirus* created by more than 500 serial passages of Chorioallantois vaccinia Ankara virus (CVA) in chicken embryo fibroblast (CEF) cells [5,6]. It was initially developed during the 1970s by Anton Mayr and colleagues in Germany to improve the safety of smallpox vaccination. MVA cannot propagate in humans or in most mammalian cells, thus reducing the risk of serious adverse events compared to replication-competent vaccines [7,8].

On 23 July 2022, the World Health Organization (WHO) declared mpox a public health emergency of international concern, which was revoked in May 2023 [9]. Pre- and post-exposure vaccination with either second-generation (ACAM2000, replicating virus vaccine) or third-generation (MVA, non-replicating, or LC16, minimally replicating) orthopox vaccines is proposed by the WHO. MVA was approved for the prevention of smallpox and mpox by the regulatory authorities in the US in 2019, and during the global mpox outbreak, it was also authorized for mpox prevention in Europe and granted emergency use authorization in children in the US. Similarly, LC16 (KM Biologics, Kumamoto, Japan) received approval for mpox prevention for all ages in Japan during 2022 [10]. Up to May 2023, 87,377 laboratory-confirmed mpox cases have been reported worldwide, with 140 deaths. Currently, the global risk is defined as moderate by the WHO [1].

We conducted a systematic review and meta-analysis of randomized-controlled trials assessing the safety and immunogenicity of the MVA vaccine. 

## 2. Materials and Methods

This review and analysis were conducted and reported according to PRISMA guidelines [11]. The protocol was registered in PROSPERO (CRD42023413224).

### 2.1. Inclusion Criteria and Outcomes

We included randomized controlled trials (RCTs) assessing individuals of all ages, with the following comparisons: MVA vs. comparator RCTs were included if they compared MVA vaccine versus no intervention, placebo, or another orthopox vaccine;Any particular MVA regimen vs. another MVA regimen-RCTs were included if they compared different dosages of MVA, formulations (liquid or lyophilized), routes of administration (subcutaneous [SC], intramuscular [IM], intradermal [ID]), and dilutions.

Trials comparing MVA vs. a comparator were included in a quantitative meta-analysis. Trials comparing strategies of vaccination among MVA-vaccinated individuals were included in a qualitative systematic review. We excluded trials comparing dryvax or ACAM2000 to interventions other than MVA. 

The primary outcome was the safety of MVA, as reflected by serious adverse events and adverse events of special interest, as defined in individual trials. Secondary outcomes included other safety outcomes: any adverse events (AEs), any cardiac AEs, local injection AEs, systemic AEs, and mortality; and immunogenicity outcomes: rates of seroconversion (as defined in individual trials) and antibody levels (measured as geometric mean titers). Since the accepted schedule for administration of MVA is two doses with an interval of 4 weeks [10], the timeframes used for safety outcome extraction were within 4 weeks from the 1st vaccine dose and >4 weeks (separated to either after the 2nd vaccine dose only or after both doses combined, according to results reported); immunogenicity outcomes were preferably extracted at 4 weeks to test for the immunogenicity of a single dose and at 6 weeks to evaluate the immunogenicity of a 2-dose regimen. While we also aimed to assess the rate of breakthrough infections, those were eventually not available from any of the trials.

### 2.2. Search Strategy and Selection Criteria

We performed a comprehensive search, regardless of language, publication status, or year of publications, in the following databases: PubMed, Cochrane Central Register of Controlled Trials (CENTRAL), and Web of Science, from inception to 28 June 2023 (last search date). For PubMed database search, we combined the term “Modified Vaccinia Ankara OR smallpox vaccine OR IMVAMUNE OR IMVANEX OR JYNNEOS” with the Cochrane filter for randomized controlled trials: ((randomized controlled trial[pt] OR controlled clinical trial[pt] OR randomized[tiab] OR placebo[tiab] OR drug therapy[sh] OR randomly[tiab] OR trial[tiab] OR groups[tiab]) NOT (animals[mh] NOT humans[mh])). For the other databases, the first search term was used and restricted to RCTs. Additional data were searched in the references of all included trials and through personal contact with the investigators of the included trials. 

### 2.3. Study Selection and Data Extraction

Two reviewers independently performed the search, applied the inclusion criteria, and conducted data extraction. Risk of bias was assessed according to the domain-based evaluation recommended by the Cochrane handbook, grading each domain as having a low, high, or unknown risk of bias [12]. The following subgroups were planned: individuals with immunodeficiency, women [13], and individuals with baseline skin disorders (e.g., atopic dermatitis), susceptible to complications of older-generation orthopox viruses. 

Sensitivity analysis was planned through allocation concealment, allocation generation, and blinding. A funnel plot to assess small-study effects was planned but not performed due to the small number of included trials [12].

### 2.4. Statistical Analysis

We calculated relative risks (RRs) and 95% confidence intervals (Cis) for each included trial. Heterogeneity was assessed using the chi-squared-based Q-test and the I^2^ measure of inconsistency [12]. Meta-analysis was performed using the Mantel–Haenszel fixed-effects model if no substantial heterogeneity was found (I^2^ ≤ 50%) and the random effect model in cases of more severe heterogeneity. For all analyses, *p* < 0.05 was considered significant. (Cochrane Review Manager (RevMan version 5.4)).

## 3. Results

The study flow chart is presented in Figure 1 (according to the PRISMA flow diagram). Twenty-six publications were retrieved for full-text assessment; 15 of them met inclusion criteria. Seven publications (eight trials) compared MVA vs. other vaccines and were included in the quantitative meta-analysis. Eight additional trials included various comparisons of MVA administration, while all patients received MVA and were included in the qualitative systematic review. 

Table 1 and Table 2 report the characteristics of the trials included in the quantitative and qualitative analyses, respectively. Among seven publications comparing MVA vs. a comparator, one compared MVA to ACAM2000 [14], while the others compared MVA with placebo. Five publications included healthy vaccine-naive individuals; two included healthy vaccine-experienced individuals [15,16]; and one included hematopoietic stem-cell transplant recipients [17]. One publication [16] included two cohorts, vaccinia naive and vaccinia experienced adults; however, results could not be extracted separately for the cohorts, and hence this trial is reported as a single cohort in our analysis. Further details on the specific comparisons, dosage, formulation, and duration of follow-up are provided in Table 1. Among the eight trials included in the systematic review part, various comparisons are detailed in Table 2, including dosage, interval between doses, formulation, lots, and way of administration. Most of the patients included were vaccine-naive; seven trials included healthy individuals, and one included HIV-positive individuals [18].

Of the seven publications included in the meta-analysis, all were considered to have a low risk of bias for both allocation generation and concealment. Six publications comparing MVA to placebo were double-blind, and the trial comparing MVA to ACAM2000 was open-label [14]. All but one trial [19] had a low risk of bias due to incomplete outcome data and selective outcome reporting. Eight of the trials were sponsored by Bavarian Nordic. For a detailed risk of bias assessment, see Appendix A.


vaccines-11-01410-t001_Table 1Table 1Characteristics of trials included in the meta-analysis (comparison: MVA vs. comparator).Study IDComparisonStudy PopulationMVA-Way of AdministrationMVA-DoseMVA-FormulationDuration of Follow-Up (Longest)No RandomizedAgePreviously VaccinatedPrimary OutcomeFrey 2007 [20]MVA (5 arms) vs. placeboHealthy, vaccinia-naiveArms 1-3: SC 2 MVA doses + DryvaxArm 4: placeboArm 5: SC 2 MVA doses + placeboArm 6: IM 2 MVA doses + DryvaxArm 1: 2 × 10^7^ TCID50Arm 2: 5 × 10^7^ TCID50Arm 3: 1 × 10^8^ TCID50Arm 5: 1 × 10^8^ TCID50Arm 6: 1 × 10^8^ TCID50Lyophilized140 days9018–32NoneAdverse events, cellular and humoral immune responses, compareroutes of administration (at days 14, 28, 42, 56, 112 and 140)Greenberg 2016 [15]MVA vs. placebo *Vaccinia-experiencedSCArm 1: 1 × 10^8^ TCID50, second dose at 4 weeksArm 2: placebo at day 0, MVA 1 × 10^8^ TCID50 at 4 wLiquidShort 8–10 w; long 6 months12056–80100%Safety in a vaccinia-experiencedpopulation after administration of either one or two doses of MVAOverton 2018 [21]MVA (3 different lots) vs. placeboHealthy, vaccinia-naiveSC1 × 10^8^ TCID50Second dose at 4 weeksLiquid26 w after second dose400518–40NoneGeometric Mean Titers (by PRNT) 2 w after second vaccinationParrino 2007 naïve [16]MVA (1, 2, or 3 doses) vs. placeboDryvax at 12 w for allHealthy, vaccinia-naiveIM1 × 10^8^ TCID50Either 1, 2 or 3 doses at week 0, 4, 12Liquid24 weeks after the lastMVA/placebo dose7618–32NoneSafety and clinical protection against vaccinia (Dryvax^®^)challengeParrino 2007 immune [16]MVA (1 or 2 doses) vs. placebo Dryvax at 12 w for allHealthy, vaccinia-immuneIM1 × 10^8^ TCID50Either 1 or 2 doses at week 0, 4Liquid24 weeks after the lastMVA/placebo dose7518–61100%Safety and clinical protection against vaccinia (Dryvax^®^)challengePittman 2019 [14]MVA vs. ACAM2000Healthy personsSC, over deltoid muscle1 × 10^8^ TCID50Second MVA dose at week 4; then ACAM2000 dose at week 8Liquid6 months after last vaccination44018–42NoneGeometric mean titers of neutralizing antibodies at 4 (ACAM) and 6 (MVA) weeksWalsh 2013 [17]MVA (2 arms of different doses) vs. placeboHSCT recipients (at least 2 years after transplant)SCArm 1: 1 × 10^7^ TCID50Arm 2: 1 × 10^8^ TCID50Second MVA dose at week 4 for both armsLiquidShort: 56 days; long: 180 d2418–60NoneSafety and reactogenicityZitzmann- Roth 2015 (and a following publication–Ilchmann 2023) [19,22]MVA vs. placebo (3 arms–MVA 2 doses, MVA one dose, placebo)Healthy, vaccinia-naiveSC1 × 10^8^ TCID50Arm 1: second MVA dose at week 4Arm 2: second dose of placebo at week 4Arm 3: 2 placebo doses, 0, 4 wLiquidShort—28–35 d after 2nd dose; long—6 m54518–55NoneZitzmann-Roth: Safety–ECG changes and cardiac symptoms. Ilchmann: Geometric mean titers of neutralizing antibodiesNC—not specified; SC—subcutaneous; HSCT—hematopoietic stem cell transplantation; PRNT—plaque reduction neutralization test. * The comparison of MVA vs. placebo was for 4 weeks, during which the placebo group received an MVA dose. For studies using Dryvax, this vaccine was administered at a later stage, and all reported outcomes collected for this meta-analysis were prior to Dryvax administration.
vaccines-11-01410-t002_Table 2Table 2Characteristics of trials included in the systematic review (all patients received MVA).Study IDComparisonStudy PopulationWay of AdministrationDoseFormulationDuration of Follow-Up (Longest)No RandomizedAgePreviously VaccinatedPrimary OutcomeFrey 2014 [23]Dose–2 arms: high dose vs. standard doseHealthy vaccinia- naïve individualsSC, over deltoid muscleHigh dose—5 × 10^8^ TCID50 single dose vs. standard dose—1 × 10^8^ TCID50 two doses (0, 28 d)LiquidReactogenicity 14 days, adverse events 28 days90Median 26.5 (range 18–37)NoneTime-to-seroconversion after the first vaccinationFrey 2015 [24]Formulation, way of administration and dose–3 arms: 1. Lypophilized SC vs. 2. Liquid SC vs. 3. Liquid IDHealthy vaccinia-naïve subjectsSC over deltoid/ID volar area of the forearmArm 1: 1 × 10^8^ TCID50 Arm 2: 1 × 10^8^ TCID50Arm 3: 2 × 10^7^ TCID50All two doses (0, 28 d)Lyophilized/liquidReactogenicity 28 days, adverse events 56 days524median 26.8 (range: 18–38)NoneGeometric mean peakvon Krempelhuber 2010 [25]Doses–3 arms: 1 × 10^8^ TCID50 vs: 2 × 10^7^ TCID50 vs: 5 × 10^7^ TCID50Healthy vaccinia-naïve subjectsSC, over deltoid muscleAll two doses (0, 28 d)Lyophilized84 days16418–30NoneDose finding in terms of safety andimmunogenicityVollmar 2006 [26]Dose, way of administration (5 groups)Healthy male subjectsArms 1–4 vaccinia-naïve, arm 5 prior vaccinationsArm 1: SCArm 2: SCArm 3: SCArm 4: IMArm 5: SCArm 1: 1 × 10^6^ TCID50Arm 2: 1 × 10^7^ TCID50Arm 3: 1 × 10^8^ TCID50Arm 4: 1 × 10^8^ TCID50Arm 5: 1 × 10^8^ TCID50Arms 1–4: two doses (0, 28 d); arm 5 one doseLiquid106 days8620–5520%safety and tolerability atdifferent dosesJackson 2017 [27]Way of administration, intervalsHealthy vaccinia-naïve subjectsSC over deltoid: Arms 1–3 by syringeand needle, arm 4 by jet injectorAll: 1 × 10^8^ TCID50Arm 1: days 1, 29Arm 2: days 1, 15Arm 3: days 1, 22Arm 4: days 1, 29LyophilizedReactogenicity 29 days after 2nd vaccine, adverse events 6 months43518–40NoneNon-inferiority of peak PRNT antibody levelsOverton 2020 [18]Dose, intervalsHIV positive, vaccinia-naïve adultsSC over deltoidArm 1: 0.5 × 10^8^ TCID50 weeks 0, 4Arm 2: 1 × 10^8^ TCID50 weeks 0, 4Arm 3: 0.5 × 10^8^ TCID50 weeks 0, 4, 12Liquid12 month8718–45NoneSerious and/or unexpected adverse eventsOverton 2023 [28]Lots (three consecutively manufactured lots of the freeze-dried MVA-BN vaccine)Healthy adultsSC over deltoid1 × 10^8^/0.5 mL dose; 2 doses, day 0, 28Lyophilized6 months112918–45NoneNeutralizing antibody immune responsesFrey 2013 [29]MVA (2 arms) vs. placebo *Healthy, vaccinia-naïveSC1 × 10^8^ TCID50Arm 1: days 0, 7Arm 2: days 0, 28Arm 3: placeboLiquidShort: 14 days after last vaccination; long: 1 y20818–35NoneGeometric mean antibody titers (PRNT) at14 days post last vaccinationSC—subcutaneous; ID—intra-dermal; PRNT—plaque reduction neutralization test. * Randomization is for vaccine schedule (0 + 7, vs. 0 + 28, 0) and not for placebo or MVA, which are mixed inside the same study arm.


### 3.1. Safety

#### 3.1.1. Following Two MVA Doses (Combined Data)

Adverse events of special interest (AESI) were reported from 3 trials (4489 participants), showing significantly higher rates in the MVA arm compared to placebo (relative risk [RR] 4.07, 95% confidence interval [CI] 1.10–15.1, without heterogeneity, Figure 2). These included any cardiac symptoms, ECG changes, or troponin elevations that were considered significant (see Appendix A for definitions in individual trials). No difference was demonstrated in SAEs (4 trials, 4513 participants, RR 0.92, 95% CI 0.48–1.77, without heterogeneity, Figure 3), any adverse events (2 trials, 144 participants, RR 0.97, 95% CI 0.88–1.06), or perimyocarditis (2 trials, 4029 participants, RR 2.32, 95% CI 0.44–12.32, without heterogeneity).

Local AEs, pain, induration, tenderness, and erythema were all significantly more common with MVA. Fever was the only systemic AE reported in one trial, showing no significant difference.

#### 3.1.2. Following a Single MVA Dose

After 1 dose of MVA, no significant difference was demonstrated for either AESI (2 trials, 795 participants, RR 0.97, 95% CI 0.5–7.79, without heterogeneity) or SAEs (4 trials, 4940 participants, RR 0.58, 95% CI 0.23–1.47, without heterogeneity); Figure 4. No difference was reported in the outcome of cardiovascular AEs (3 trials, 350 participants, RR 1.75, 95% CI 0.52–5.80, I^2^-27%). Two trials (4445 participants) reported AEs requiring discontinuation of the vaccination schedule without significant difference for this outcome (RR 2.3, 95% CI 0.75–7.03, without heterogeneity).

Overton et al., including 4005 participants, reported overall higher rates of AEs with MVA compared with placebo (RR 1.79, 95% CI 1.67–1.91) [21]. These were mainly injection-site AEs.

Local AEs reported in general, and specifically injection site pain, tenderness, and induration, were significantly more common with MVA compared with placebo; however, significantly more common with ACAM2000 compared with MVA in one trial. (14) Systemic adverse events were also more common with ACAM2000 compared to MVA. MVA compared to placebo did not result in significantly more common systemic adverse events (1 trial), fever (2 trials), fatigue (2 trials), or gastrointestinal adverse events (1 trial). Arthralgia/myalgia were significantly more common with MVA compared to placebo (2 trials). One case of mortality was reported with MVA (1/3003 patients) versus none (0/1002) in the placebo group. This case of suicide was assessed as unrelated to the study treatment by the investigator [21].

#### 3.1.3. Following Second MVA Dose Separately

Few studies reported safety outcomes separately after the second dose of the vaccine. Two trials reported higher rates of local AEs with MVA [15,20], one trial reported higher rates of systemic AEs [20], and one trial reported higher rates of any adverse events with MVA [21]. SAEs were reported from one trial with no difference [21], and no mortality cases were reported in any of the included trials. 

### 3.2. Immunogenicity

Seroconversion by plaque reduction neutralization test (PRNT) after 2 MVA doses was significantly higher compared with placebo in 3 trials including vaccine-naïve participants (RR 33.47, 95% CI 12.97–90.58, with substantial heterogeneity, I^2^ = 72%). Compared with ACAM2000 [14], MVA had a trend for higher seroconversion rates by PRNT (RR 1.03, 95% CI 1.0–1.06) (Figure 5).

Seroconversion rates by ELISA were similarly significantly higher with MVA vs. placebo in two trials (RR 38.88, 95% CI 21.28–71.05, I^2^ = 40%), as well as higher compared to ACAM2000 in a single trial (RR 1.04, 95% 1.01–1.07).

Seroconversion rates by both methods were not significantly different between subjects who received two doses of MVA and placebo followed by one dose of MVA in vaccine-experienced participants [15].

Following a single MVA dose, higher seroconversion rates by PRNT were demonstrated in one trial compared with placebo in vaccine-naïve patients [19] and similar rates to ACAM2000 [14]. Similarly, using ELISA, one trial showed higher seroconversion rates versus placebo in vaccine naïve [19] and one in vaccine experienced [15]; there was no difference compared to ACAM2000 [14].

Antibodies with geometric mean titers (GMT) were reported in a few studies. Greenberg et al. reported no difference in GMT between MVA and placebo 2 weeks after the first dose among vaccine-experienced participants (by PRNT mean difference [MD] −12.3, 95 % CI −86.2 to 61.7, by ELISA −16.7, 95% CI −282.6 to 316) [15]. Overton et al. demonstrated significantly higher GMT by ELISA at 2 weeks after the first dose among vaccine-naïve participants (MD 877.8, 95% CI 843.5–912) [21].

Pittman et al. demonstrated significantly higher GMT with ACAM2000 compared to MVA at 4 weeks (single dose) (PRNT, MD −62.4, 95% CI −76.2 to −48.6, ELISA MD −64.8, 95% CI −108 to −21.6). However, at 6 weeks (2 doses), antibody levels were significantly higher with MVA (PRNT, MD 88.8, 95% CI 65.6–112, ELISA MD 927.7, 95% CI 797–1058.4) [14].

### 3.3. Comparison of Different MVA Regimens

We identified eight RCTs (2723 participants) comparing different MVA vaccine regimens (administration routes and dosages) (Table 2). Seven trials included primarily vaccine-naïve healthy adults, while one included vaccine-naïve HIV carriers [18].

Four of the eight RCTs (50%) compared different vaccine dosages. A higher MVA dose (1 × 10^8^ TCID_50_) was more effective than lower doses, as reflected by both seroconversion and total antibody response, without compromising safety, in phase I [26] and phase II [25] studies. Additionally, a single high-dose (1 × 10^8^ TCID_50_) MVA vaccine is unlikely to provide a significant benefit over two standard doses [23]. Among vaccine-naïve HIV carriers, a double dose (twice the number of injections on each administration day) was found to be as safe as a standard dose (≥0.5 × 10^8^ TCID_50_). However, doubling the dose or adding a third booster was deemed unnecessary in terms of immunogenicity, even in this population of immunocompromised individuals [18].

Two of the eight RCTs (25%) assessed vaccination intervals between the first and second vaccine doses, with 28 days as the standard interval. A compressed dosing interval of either 7 [29], 15, or 22 [27] days was inferior to the standard interval in terms of antibody responses, concluding that the standard vaccination interval is preferred.

Three of the eight (38%) RCTs assessed vaccine administration routes. Non-naïve individuals (i.e., those previously vaccinated for smallpox) were more likely to develop local reactions, likely reflecting an accelerated booster effect resulting from pre-existing immunity. Pain reactions were slightly more intense in the intramuscular than in the subcutaneous route [26]. Subcutaneous and intradermal administration resulted in similar antibody responses; however, local reactions were significantly more frequent following intradermal administration, although none were severe enough to conclude that this administration route should be totally avoided [24]. Subcutaneous administration by jet injector was non-inferior to syringe and needle in terms of immunogenicity; however, the former carried a higher rate of injection site reactions [27].

A single RCT (13%) assessed three consecutive manufactured lots of MVA vaccine, all of which led to a consistent immunogenicity and safety profile [28].

## 4. Discussion

In this systematic review and meta-analysis of RCTs, we found that cardiovascular AEs (reported as AESI) following two MVA doses were significantly more common compared to placebo, though SAEs, peri-myocarditis cases, and other systemic AEs were not significantly different. Specifically, following a single dose of MVA, no difference compared with placebo was demonstrated for the outcomes of SAEs, AESI, AEs requiring discontinuation, or peri-myocarditis. These findings provide evidence for the safety of any regimen of the vaccine, and specifically of a single dose. Additional studies should evaluate cardiovascular AEs specifically among older adults, considering the 8% of cardiovascular AEs demonstrated in the vaccine arm in the study by Greenberg et al. (Figure 2) [15]. Local AEs were more common with MVA versus placebo, though significantly less common compared to ACAM2000. Data on adverse events specifically following the second dose were limited, though SAEs were not more common. Despite the variations in vaccine dosing, formulation, way of administration, and different trial designs, no substantial heterogeneity has been demonstrated for any of the safety outcomes. 

Seroconversion rates were significantly higher compared with placebo after a single or two doses. Compared with ACAM2000, one MVA dose achieved similar seroconversion rates, with a trend for higher rates after two doses. The substantial heterogeneity demonstrated for the immunogenicity outcomes stemmed from the magnitude of the effect rather than the direction (i.e., all trials demonstrated higher immunogenicity with MVA, but the size of the effect differed). 

The immunological persistence of the vaccine was tested in a study reported by Ilchmann et al. [22] In this study, 152 participants received an MVA booster two years after having been primed with one or two doses of MVA. Neutralizing antibodies increased 2 weeks after the booster dose from 1.1 to 80.7 in those who previously received one MVA dose and from 1.3 to 125.3 in those who previously received two doses, values that were higher than following the primary vaccination. At a six-month follow-up, GMT were 25.6 and 49.3 for those with previous one and two doses, respectively. There were no safety concerns with this booster dose [19,22].

Our results on the effectiveness and safety of two doses of MVA vaccine reinforce the 2019 United States’ Advisory Committee on Immunization Practices (ACIP) recommendation on MVA (JYNNEOS) for preexposure prophylaxis for populations at risk for orthopoxviruses. A particular advantage of JYNNEOS is its safety in several circumstances in which ACAM2000 is contraindicated: during pregnancy or among individuals with immune suppression or dermal comorbidities [30].

None of the RCTs evaluated the clinical effectiveness of the vaccine in preventing mpox.

The effectiveness of MVA vaccine against mpox was recently evaluated in three case-control studies from the US that reported an adjusted effectiveness ranging from 36–86% for 1 dose and from 66 to 89% for 2 doses, implying that patients vaccinated with any regimen are less likely to have mpox, with a lower probability after two vaccine doses [31,32,33]. Higher effectiveness (76–89% for two doses, 41–72% for one dose) was demonstrated for patients without immunocompromising conditions [31,32]. These results are supported by additional data from the Centers for Disease Control and Prevention (CDC), which evaluated 9544 mpox cases among men aged 19–49. The incidence rate ratio of mpox among unvaccinated men was 7.4 compared to one vaccine dose and 9.6 compared to two doses, supporting vaccine effectiveness with the larger effect of two doses. This study also demonstrated no difference between intradermal and subcutaneous administration [34].

A small, real-life effectiveness study from Israel evaluated a single subcutaneous dose of MVA among men with HIV or receiving PrEP. This study showed adjusted vaccine effectiveness of 86% at a follow-up of an average ~5 months [35]. Similar rates (78%) of vaccine effectiveness were also demonstrated for a single MVA dose in a UK study, at an average follow-up of ~2.5 months [36].

A recent meta-analysis evaluated the safety of MVA among vaccinia-naïve versus vaccinia-experienced people and demonstrated higher rates of adverse events in the former [37]. The prescribing information for MVA (JYNNEOS) summarizes data from 22 studies (7859 individuals) and reports some variability in AEs between vaccine-experienced and naïve individuals, along with similar rates of AEs between healthy individuals and those with HIV or atopic dermatitis [38].

The main limitation of our meta-analysis is the lack of RCTs assessing clinical outcomes. This is particularly relevant as none of the included trials were carried out during the recent outbreak, in which the clinical presentation of mpox differed from what was known before [39]. Though antibody titers are used as surrogates for vaccine effectiveness, prevention of infection is the main aim of vaccination. 

Moreover, during the recent outbreak, severe illness and mortality from mpox were uncommon, except for individuals with immunocompromising conditions, particularly those with HIV and AIDS. These populations were underrepresented in the included RCTs. Of note, there is additional data on the immunogenicity and safety of MVA in HIV-positive individuals from two clinical trials that were not included in this meta-analysis as they were not randomized [40,41]. Further comparative studies assessing effectiveness are necessary to determine the optimal vaccination regimen for these populations. In addition, the design of the included trials was heterogenous, the evaluated population differed, various vaccine regimens were used, and the number of included trials was limited. Though these did not affect the heterogeneity of our results, additional studies are needed in order to provide recommendations regarding the optimal vaccine regimen. The study by Overton et al. included 4005 patients, a substantially larger number compared to other studies. Nevertheless, as mentioned above, no heterogeneity between trials’ results was demonstrated. 

In summary, our meta-analysis provides reassuring results concerning the immunogenicity and safety of one or two doses of the MVA vaccine. This is in line with clinical data showing considerable vaccine effectiveness with one MVA dose, with higher effectiveness with two doses. Further studies are needed to evaluate the immunogenicity of a single dose; future studies should also address clinical outcomes after one and two vaccine doses. A single dose may be considered according to vaccine availability, with preference for two doses.

## Figures and Tables

**Figure 1 vaccines-11-01410-f001:**
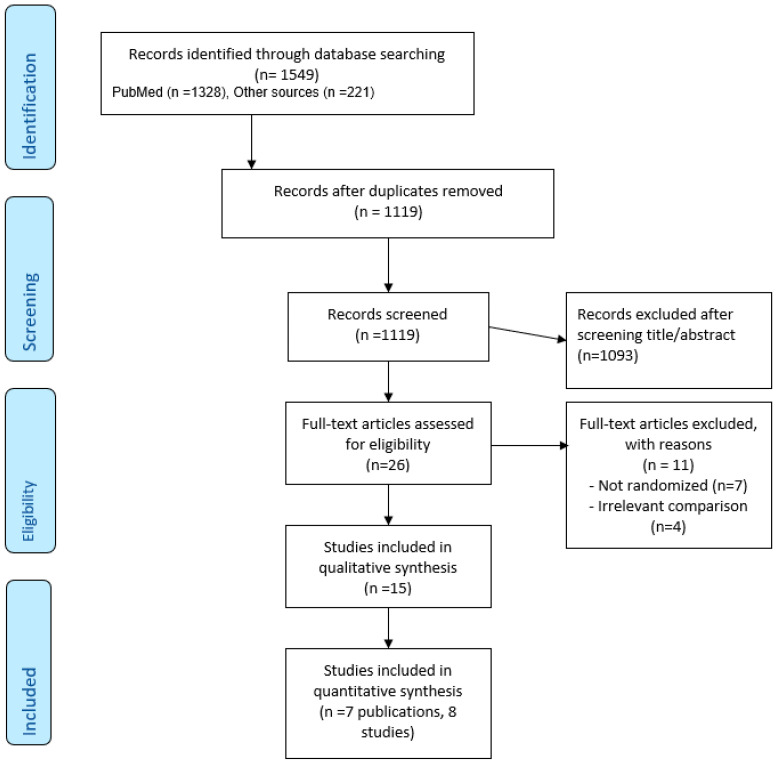
Study flowchart.

**Figure 2 vaccines-11-01410-f002:**
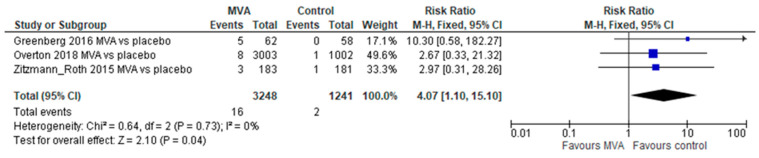
Adverse events of special interest following both MVA doses.

**Figure 3 vaccines-11-01410-f003:**
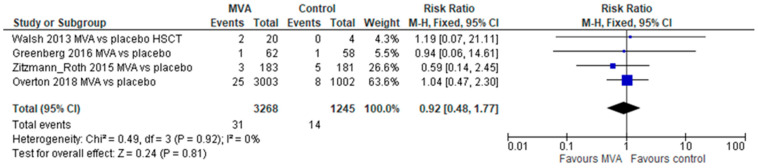
Serious adverse events following both MVA doses.

**Figure 4 vaccines-11-01410-f004:**
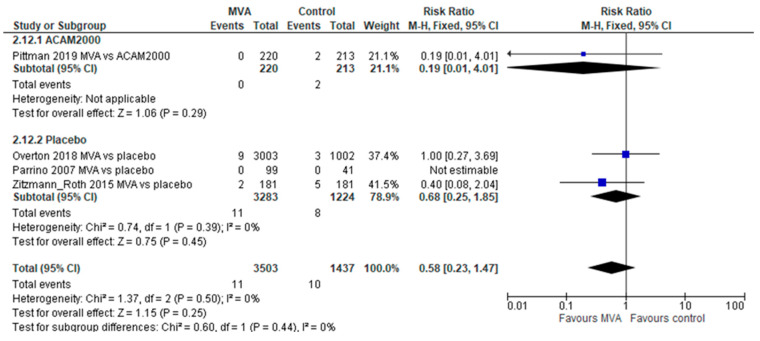
Serious adverse events following single-dose MVA.

**Figure 5 vaccines-11-01410-f005:**
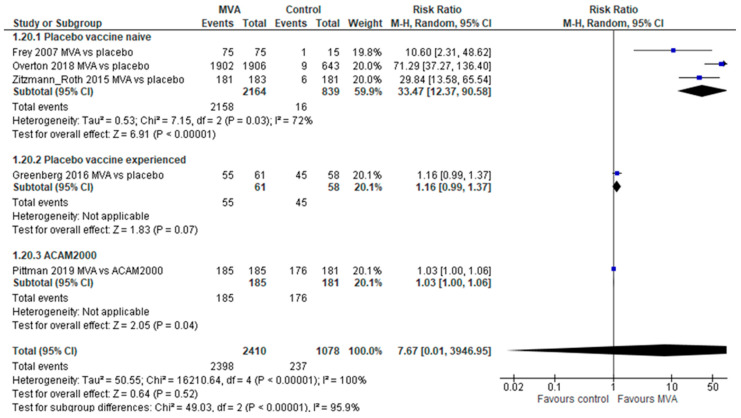
Seroconversion rates by plaque reduction neutralization test (PRNA) following two vaccine doses.

## Data Availability

Data can be shared upon reasonable request from the corresponding author.

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
