# Peer review of "Immunogenicity and Safety of Modified Vaccinia Ankara (MVA) Vaccine—A Systematic Review and Meta-Analysis of Randomized Controlled Trials"

_vaccines, 2023, doi:10.3390/vaccines11091410_

Round 1

Reviewer 1 Report

Estimated Authors,

I've read with great interest your systematic review + metanalysis entitled "Immunogenicity and safety of Modified Vaccinia Ankara (MVA) vaccine – a systematic review and meta-analysis of randomized controlled trials".

In this study, data from published studies and RCT were pooled in order to provide a series of information about the actual immunogenicity and safety of MVA compared either to placebo or other available vaccines.

Because of the raising interest on mpox, the present paper is quite interesting for international readers, but will require substantial editing before its eventual acceptance, from several point of view.

From a general point of view, I've noticed the following shortcomings:

1. the definition of AESI is, in fact, limited to cardiovascular events. This should be more clearly reported and discussed. Moreover, as AESI are mostly cardiovascular events, a more appropriate reporting should report pooled estimates for: adverse events, cardiovascular events, AESI. Moreover, I would expect a summary of reported adverse events, where available.

2. albeit dosage are quite consistent across studies, their design is on the contrary quite heterogenous. Authors have properly managed potential heterogeneity via a proper meta-analysis model, but this did reasonably affect pooled estimates, despite the extensive and again appropriate use of sub-group analyses. Please discuss more extensively.

3. Results section is quite descriptive, shifting from the general analysis of results to the reporting of individual characteristics of a single study that resembles a kind of discussion. In other words, results section should be extensively rewritten with a more systematic approach, moving some of individual comments to the discussion section.

4. Discussion section, by acknowledging the aforementioned limits on design of adverse events, should be more cautious in addressing the actual safety of this specific vaccine.

5. The individual size of the study of Overton et al. impacts on all your estimates, and this specific and very significant shortcoming must be preventively and extensively acknowledged across the discussion. On the other hand, I'm quite doubtful about the accuracy of a fixed model analysis for seroconversion rates (see figure 5: Overton with 1906 out of 2410 samples is weighted around 20%, compared to the 79.1% of actual weight!).

The English text is clearly appropriate in terms of grammar: no main issue were detected. On the other hand, it lacks proper flows, and an accurate revision aimed to improving the readability is recommended.

Author Response

Estimated Authors,

I've read with great interest your systematic review + metanalysis entitled "Immunogenicity and safety of Modified Vaccinia Ankara (MVA) vaccine – a systematic review and meta-analysis of randomized controlled trials".

In this study, data from published studies and RCT were pooled in order to provide a series of information about the actual immunogenicity and safety of MVA compared either to placebo or other available vaccines.

Because of the raising interest on mpox, the present paper is quite interesting for international readers, but will require substantial editing before its eventual acceptance, from several point of view.

From a general point of view, I've noticed the following shortcomings:

  1. the definition of AESI is, in fact, limited to cardiovascular events. This should be more clearly reported and discussed. Moreover, as AESI are mostly cardiovascular events, a more appropriate reporting should report pooled estimates for: adverse events, cardiovascular events, AESI. Moreover, I would expect a summary of reported adverse events, where available.

Answer: As mentioned in the methods section, we extracted “adverse events with special interest, as defined in individual trials”. Indeed, practically, as we elaborated in Supplemental Table 1, AESI were solely significant cardiac events. We added to the results section, under safety - AESI: “These included any cardiac symptoms, ECG changes or troponin elevations that were considered significant”. To clarify further – we revised the ‘cardiovascular AE’ reported from two trials, to ‘peri-myocarditis’ (that is specifically reported in these studies). Any AEs rates and SAEs rates are also reported in the results. Subsequently, we also revised the relevant discussion section to: “In this systematic review and meta-analysis of RCTs, we found that cardiovascular AEs (reported as AESI) following two MVA doses were significantly more common compared to placebo, though SAEs, peri-myocarditis and other systemic AEs were not significantly different. Specifically following a single dose of MVA, no difference compared with placebo was demonstrated for the outcomes of SAEs, AESI, AEs requiring discontinuation, or peri-myocarditis.”. The abstract was also revised to change from AESI to “We found that cardiovascular adverse events following two MVA doses were significantly more common”

  1. albeit dosage are quite consistent across studies, their design is on the contrary quite heterogenous. Authors have properly managed potential heterogeneity via a proper meta-analysis model, but this did reasonably affect pooled estimates, despite the extensive and again appropriate use of sub-group analyses. Please discuss more extensively.

Answer: We agree with the above. As mentioned by the reviewer, in terms of methods we have applied the appropriate measures, hence we could only add text clarifying and emphasizing this limitation. We added to the discussion section: - to the paragraph reporting safety outcomes: “Despite variations in vaccine dosing, formulation, way of administration and the different trial designs, no substantial heterogeneity has been demonstrated for any of the safety outcomes.”; to the limitation paragraph: “In addition, the design of the included trials was heterogenous, evaluated population differed, various vaccine regimens were used, and the number of included trials was limited. Though these did not affect heterogeneity of our results, additional studies are needed in order to provide recommendations regarding the optimal vaccine regimen.

  1. Results section is quite descriptive, shifting from the general analysis of results to the reporting of individual characteristics of a single study that resembles a kind of discussion. In other words, results section should be extensively rewritten with a more systematic approach, moving some of individual comments to the discussion section.

Answer: The results section has been extensively revised, with some parts deleted or moved to the discussion.

  1. Discussion section, by acknowledging the aforementioned limits on design of adverse events, should be more cautious in addressing the actual safety of this specific vaccine.

Answer: As detailed in the answers to the above comments, we revised the discussion section, including the limitations section.

  1. The individual size of the study of Overton et al. impacts on all your estimates, and this specific and very significant shortcoming must be preventively and extensively acknowledged across the discussion. On the other hand, I'm quite doubtful about the accuracy of a fixed model analysis for seroconversion rates (see figure 5: Overton with 1906 out of 2410 samples is weighted around 20%, compared to the 79.1% of actual weight!).

Answer: We agree that this should be mentioned, and we added this to the limitation section. Nevertheless, the results of this study for safety outcomes were in line with other studies, with no heterogeneity between results. Heterogeneity of immunogenicity outcomes, as reported in the manuscript, was caused due to different magnitudes of the effect in the same direction of the graph. Regarding the weight – a weight of a study in RevMan is determined considering both the nominators and denominators of the results, hence it can vary and even if the size of the study is large, the weight can be modest if the nominator in small. 

Comments on the Quality of English Language

The English text is clearly appropriate in terms of grammar: no main issue were detected. On the other hand, it lacks proper flows, and an accurate revision aimed to improving the readability is recommended.

Answer: Thank you. We tried to shorten the text and create a better flow.

Reviewer 2 Report

Vaccines- Peer review summary 08.07.2023

 Systematic Review 1

Immunogenicity and safety of Modified Vaccinia Ankara 2 (MVA) vaccine – a systematic review and meta-analysis of randomized controlled trials

Background

An overall well written manuscript from a distinguished Israeli group and one member of the company that produced the vaccine MVA provided in the USA under the name JYNNEOS® .

The manuscript reports a summary of important data related primarily to the vaccine specific and general adverse effects of the MVA vaccine as extracted from the literature.

MPox is a relatively esoteric condition to most noninfectious disease clinicians such as the reviewer and as such this manuscript serves to provide important insight and education related to vaccines for MPox.

Strengths

a.     The authors used the PRISMA guidelines for systematic reviews which is the standardfor reporting. The study/ manuscript was registered as a PROSPERO study and verified by the reviewer.

b.     Careful and critical scrutiny of all published studies globally related to the MVA vaccine.

c.      Concise description of earlier vaccines related to their adverse event history.

d.     Extensive description on the inclusion criteria related to the existing studies.

e.      Clear description of the search strategy, study selection and statistical analysis.

f.       A focus on the important safety outcomes as expressed as AEs, SAEs and specific interest AEs (AESI). These are critical as use of this replication attenuated viral vector would give concern to both medical practitioners and the general public globally.

g.     Dr Leinert expertise as a pharmaceutical scientist in the company Bavarian Nordic AG, Zug, Switzerland the company that has developed the vaccine from the web site as - JYNNEOS® / IMVAMUNE® / IMVANEX®

h.     An excellent complete summary description of the studies the group were able to evaluate which was quite small but important.

The reviewer did access the PI data from the approved JYNNEOS®. As per the PI- “The overall clinical trial program included 22 studies and a total of 7,859 individuals 18 through 80 years of age who received at least 1 dose of JYNNEOS (7,093 smallpox vaccine-naïve and 766 smallpox vaccine-experienced individuals).”

 Question- would it be relevant for the authors' to summarize these findings vis a vis the current reported studies accessed in this review? A clinician would want to know based on the PI reported trials how they are same or different than the trials assessed in this manuscript.

The reviewer also accessed the MMWR report through PUBMED

Use of JYNNEOS (Smallpox and Monkeypox Vaccine, Live, Nonreplicating) for Preexposure Vaccination of Persons at Risk for Occupational Exposure to Orthopoxviruses: Recommendations of the Advisory Committee on Immunization Practices - United States, 2022. MMWR Morb Mortal Wkly Rep. 2022 Jun 3;71(22):734-742. doi: 10.15585/mmwr.mm7122e1. Erratum in: MMWR Morb Mortal Wkly Rep. 2022 Jul 08;71(27):886. PMID: 35653347; PMCID: PMC9169520.

A very robust summary of findings to date.

Question- Are there some findings in the USA review that would be useful to describe in your manuscript?

Results-

1.     The data is very reassuring in both AEs and AESI that has been commented on in other reviews and approvals. However, the dominant study by Overton et al with over 3000 subjects dwarfs the other noted studies. Any concerns related to this finding in verifying outcome validity?

2.     Fig 2 AESI- It appears the small study of Greenberg et al has a high number compared to the other reported studies however the SAEs in Fig 3 are essentially identical. Any concerns about the AESI reporting from the Greenberg et al study? Perhaps an age difference in the studies?

Overall, an excellent article summarizing the “real world” experience in trials conducted focused on the safety concerns and the surrogate immunological responses. The manuscript highlights the fact that international travel has broken down the infectious disease silos in which diseases like MPox were confined.

As the authors have stated there is no robust clinical vaccine efficacy outcomes to date but based on safety and surrogate immunological results it is hopeful this vaccine will be effective.

Author Response

Systematic Review 1

Immunogenicity and safety of Modified Vaccinia Ankara (MVA) vaccine – a systematic review and meta-analysis of randomized controlled trials

Background

An overall well written manuscript from a distinguished Israeli group and one member of the company that produced the vaccine MVA provided in the USA under the name JYNNEOS® .

The manuscript reports a summary of important data related primarily to the vaccine specific and general adverse effects of the MVA vaccine as extracted from the literature.

MPox is a relatively esoteric condition to most noninfectious disease clinicians such as the reviewer and as such this manuscript serves to provide important insight and education related to vaccines for MPox.

Strengths

  1. The authors used the PRISMA guidelines for systematic reviews which is the standardfor reporting. The study/ manuscript was registered as a PROSPERO study and verified by the reviewer.
  2. Careful and critical scrutiny of all published studies globally related to the MVA vaccine.
  3. Concise description of earlier vaccines related to their adverse event history.
  4. Extensive description on the inclusion criteria related to the existing studies.
  5. Clear description of the search strategy, study selection and statistical analysis.
  6. A focus on the important safety outcomes as expressed as AEs, SAEs and specific interest AEs (AESI). These are critical as use of this replication attenuated viral vector would give concern to both medical practitioners and the general public globally.
  7. Dr Leinert expertise as a pharmaceutical scientist in the company Bavarian Nordic AG, Zug, Switzerland the company that has developed the vaccine from the web site as - JYNNEOS®IMVAMUNE® / IMVANEX®
  8. An excellent complete summary description of the studies the group were able to evaluate which was quite small but important.

 We thank the reviewer for his support of our manuscript and the kind words.

The reviewer did access the PI data from the approved JYNNEOS®. As per the PI- “The overall clinical trial program included 22 studies and a total of 7,859 individuals 18 through 80 years of age who received at least 1 dose of JYNNEOS (7,093 smallpox vaccine-naïve and 766 smallpox vaccine-experienced individuals).”

 Question- would it be relevant for the authors' to summarize these findings vis a vis the current reported studies accessed in this review? A clinician would want to know based on the PI reported trials how they are same or different than the trials assessed in this manuscript.

 Answer: We thank the reviewer for his meticulous search. We referred to the PI mainly regarding AEs in the discussion (we added the following paragraph: “The prescribing information of MVA (JYNNEOS) summarizes data from 22 studies (7,859 individuals), and reports some variability in AEs between vaccine experienced and naïve individuals; along with similar rates of AEs between healthy individuals and those with HIV or atopic dermatitis”). Regarding other outcomes – the PI mentions indeed 22 studies. However, several of these studies were not randomized controlled trials and therefore did not meet the inclusion criteria for this systematic review and meta-analysis.

The reviewer also accessed the MMWR report through PUBMED

Use of JYNNEOS (Smallpox and Monkeypox Vaccine, Live, Nonreplicating) for Preexposure Vaccination of Persons at Risk for Occupational Exposure to Orthopoxviruses: Recommendations of the Advisory Committee on Immunization Practices - United States, 2022. MMWR Morb Mortal Wkly Rep. 2022 Jun 3;71(22):734-742. doi: 10.15585/mmwr.mm7122e1. Erratum in: MMWR Morb Mortal Wkly Rep. 2022 Jul 08;71(27):886. PMID: 35653347; PMCID: PMC9169520.

A very robust summary of findings to date.

Question- Are there some findings in the USA review that would be useful to describe in your manuscript?

Answer: We thank the reviewer for referring us to the MMRW report on MVA vaccine. The scientific publications that lay the foundations for the adoption of the MVA vaccine as an alternative for ACAM2000 were retrieved through our systematic review and those who met the inclusion criteria were included in our meta-analysis (references 30-33, 37).

As suggested by the reviewer, we have now referred the readers to this important report, while emphasizing that ACIP recommended the use of MVA vaccine as an alternative to ACAM2000, considering its effectiveness and safety as well as its benefits in circumstances in which ACAM2000 is contraindicated.

Results-

  1. The data is very reassuring in both AEs and AESI that has been commented on in other reviews and approvals. However, the dominant study by Overton et al with over 3000 subjects dwarfs the other noted studies. Any concerns related to this finding in verifying outcome validity?

Answer: This was also commented by reviewer 1 and we added this notification to the limitations section. Nevertheless, the safety outcomes were all without heterogeneity, pointing to similar results from all studies.

  1. Fig 2 AESI- It appears the small study of Greenberg et al has a high number compared to the other reported studies however the SAEs in Fig 3 are essentially identical. Any concerns about the AESI reporting from the Greenberg et al study? Perhaps an age difference in the studies?

Answer: Participants in the Greenberg study were indeed older from those in other studies (age 56-80). Hence, we added to the discussion: “Additional studies should evaluate cardiovascular AEs specifically among older adults, considering the 8% of cardiovascular AEs demonstrated in the vaccine arm in the study by Greenberg et al. (Figure 2)”. Regarding the difference from SAEs, we think this is reassuring, since the definition of SAEs in clinical trials is more robust than the subjective definition of AESI, specific for the MVA trials, of “Any cardiac symptoms and/or ECG changes determined to be clinically significant or troponin I elevations >X2 upper limit of normal”.

Overall, an excellent article summarizing the “real world” experience in trials conducted focused on the safety concerns and the surrogate immunological responses. The manuscript highlights the fact that international travel has broken down the infectious disease silos in which diseases like MPox were confined.

As the authors have stated there is no robust clinical vaccine efficacy outcomes to date but based on safety and surrogate immunological results it is hopeful this vaccine will be effective.

Reviewer 3 Report

In this manuscript titled "Immunogenicity and safety of Modified Vaccinia Ankara (MVA) vaccine – a systematic review and meta-analysis of randomized controlled trials", the authors did a thorough systemic review on MVA and provided important insights on whether two doses of MVA are safe and whether one dose or two doses is a better vaccination schedule for MVA. Overall, this is a well-performed analysis. I hope the following minor comments can help the authors to further improve the quality of this manuscript.

Minor issues:

1. Line 52, LC16 needs to be introduced.

2. Line 167-168, "Local AEs, pain, induration, tenderness, and erythema were all significantly more common with MVA (data not shown)." If the authors want to make these claims, data have to be shown. If the authors don't want to show the related analysis data, then these claims need to be deleted.

3. Line 199, "3.1.3. Following second MVA dose", this title is a little confusing. In this small section, are the authors comparing the AEs after second dose to the AEs after the first dose? Or are they comparing the AEs only after second dose to the AEs only after second placebo? A clearer title is needed here.

Author Response

Comments and Suggestions for Authors

In this manuscript titled "Immunogenicity and safety of Modified Vaccinia Ankara (MVA) vaccine – a systematic review and meta-analysis of randomized controlled trials", the authors did a thorough systemic review on MVA and provided important insights on whether two doses of MVA are safe and whether one dose or two doses is a better vaccination schedule for MVA. Overall, this is a well-performed analysis. I hope the following minor comments can help the authors to further improve the quality of this manuscript.

Minor issues:

  1. Line 52, LC16 needs to be introduced.

Answer: we added that LC16 is a minimally replicating virus vaccines, manufactured by KM Biologics, Japan.

  1. Line 167-168, "Local AEs, pain, induration, tenderness, and erythema were all significantly more common with MVA (data not shown)." If the authors want to make these claims, data have to be shown. If the authors don't want to show the related analysis data, then these claims need to be deleted.

Answer: The data was added as a figure S2.

  1. Line 199, "3.1.3. Following second MVA dose", this title is a little confusing. In this small section, are the authors comparing the AEs after second dose to the AEs after the first dose? Or are they comparing the AEs only after second dose to the AEs only after second placebo? A clearer title is needed here.

Answer: We agree that this is confusing and we refer to the second option listed above. We tried to clarify revising to: “Following second MVA dose separately:”, hoping this is clearer.

Round 2

Reviewer 1 Report

My previous concerns have been properly addressed.

I've no further requests.